# A Modified Corona Score Using Lung Ultrasound to Identify COVID-19 Patients

**DOI:** 10.3390/diagnostics14010051

**Published:** 2023-12-26

**Authors:** Costantino Caroselli, Michael Blaivas, Yale Tung Chen, Matteo Marcosignori, Antonio Cherubini, Daniele Longo

**Affiliations:** 1Acute Geriatric Unit, Geriatric Emergency Room and Aging Research Centre IRCCS INRCA, 60127 Ancona, Italy; costantinocost@yahoo.it; 2Department of Medicine, School of Medicine, University of South Carolina, Columbia, SC 29209, USA; mike@blaivas.org; 3Internal Medicine Department, Hospital Universitario Puerta de Hierro, 28222 Majadahonda, Spain; yale.tung.chen@gmail.com; 4Emergency Department, Azienda Ospedaliero Universitaria, Ospedali Riuniti, 60123 Ancona, Italy; matteo.marcosignori@ospedaliriuniti.marche.it; 5Department of Prevention, APSS, 38123 Trento, Italy; daniele.longo@apss.tn.it; 6Department of Diagnostics and Public Health, School of Medicine and Surgery, Università di Verona, 37124 Verona, Italy

**Keywords:** lung ultrasound, point of care, SARS-CoV-2, COVID-19, corona score

## Abstract

Background: COVID-19 continues to circulate around the world with multiple different strains being active at once. While diagnosis with antigen and molecular testing is more readily available, there is still room for alternative methods of diagnosis, particularly in out-of-hospital settings, e.g., home or nursing homes, and in low–medium income countries, where testing may not be readily available. Study Objectives: To evaluate the performance of two modified corona score methods compared with a traditional corona score approach to identify patients with COVID-19. Methods: This was a retrospective multicenter study performed to compare the ability to predict SARS-CoV-2 test results on a nasopharyngeal swab between the corona scores and two novel corona scores (modified 1 corona score (M1CS) and modified 2 corona score (M2CS)). The M1CS included lung ultrasound (LUS) and chest X-ray (CXR) results, while the M2SC only utilized LUS findings without CXRs. Emergency physicians performed point-of-care LUS and a physical examination upon admission to the emergency department. Results: Subjects positive for SARS-CoV-2 were older and had higher ferritin levels and temperature and lower diastolic blood pressure and oxygen saturation. The two groups differed on corona score and modified corona scores (*p* < 0.001 for all). SARS-CoV-2-positive patients had fewer pleural line irregularities (*p* = 0.025) but presented more frequently with an interstitial pattern on CXRs (*p* < 0.001). Conclusions: In our study, LUS alone provided a valuable contribution to the corona score and improved its performance more than when CXR results were included. These results suggest that resource-limited areas where CXRs may be unavailable or prohibitively expensive can utilize an ultrasound as the sole imaging modality without a loss of diagnostic performance for SARS-CoV-2 pneumonia diagnosis.

## 1. Introduction

The majority of emergency departments (Eds) have stopped universal testing of admitted patients with COVID-19 nasal swabbing, potentially making the diagnosis of COVID-19 more challenging due to the variety and frequent lack of specific symptoms [1,2]. 

During the worst period of the pandemic, a crucial objective was to identify a method to accurately and quickly diagnose SARS-CoV-2 infection, especially in remote areas where it is difficult to have nasal swab results available in real-time for SARS-CoV-2 detection or second-level instrumental investigations, such as HRCT. For these reasons, risk scores were proposed using alternative and cheaper methods more readily available in community hospitals, low-intensity healthcare facilities, residences, and rural and remote areas. 

The Idea of using risk scores to identify SARS-CoV-2 infection early is important not only for economic efficiency but even more to avoid unnecessary hospital admissions, reducing the waiting times during peak SARS-CoV-2 case and testing periods. The objective of scoring systems is to derive a result that can direct the most precise path to a negative or positive SARS-CoV-2 diagnosis [3]. For this reason, it is necessary to create scoring systems that are well-standardized, in which inter-operator variability is reduced as much as possible.

Among these scores was an easy-to-use algorithm, called coronascore, which was used to identify patients suspected of COVID-19 infection at the emergency department with respiratory symptoms [4,5,6]. In comparison, a chest X-ray has been demonstrated to have a low sensitivity in the diagnostics of SARS-CoV-2 infection. In fact, relying solely on chest X-ray imaging results in approximately 40% of false-negative diagnoses [7]. Chest computed tomography (CT) is a sensitive tool that is commonly used for the diagnosis and monitoring of the evolution of SARS-CoV-2-related pneumonia, but its use is limited by a lack of ready availability, risk related to radiation exposure, and a requirement for patient transport outside the ED. 

During the pandemic, the use of lung ultrasound (LUS) has been suggested as a valid alternative to chest X-rays (CXRs) and chest CT to diagnose COVID-19 pneumonia. LUS has multiple advantages, including that it can be used at the patient’s bedside, does not require patient transfer to the CT suite, can be performed rapidly, is easily repeatable, does not utilize ionizing radiation, and is easily learned by medical providers [8]. LUS has been previously demonstrated to have high sensitivity for diagnosing COVID-19 pneumonia [9]. 

A recent prospective study of 152 patients showed a significant relationship between high-resolution chest computed tomography (HRCT) and LUS. Importantly, no positive findings on computed tomography were missed on lung ultrasound [10]. Additionally, when different scenarios are examined, the severity of LUS findings appears to be associated with clinical outcomes [11,12]. 

The aim of this study is to analyze, retrospectively, the diagnostic accuracy of a novel corona score, including the use of an LUS examination instead of chest X-rays, in patients admitted to the ED due to COVID-19.

## 2. Materials and Methods

### 2.1. Study Design

This is a retrospective multicenter study of patients admitted to the emergency department (ED) in Madrid (Spain), an ED in Ancona (Italy), and the Acute Geriatric Unit of Ancona (Italy). This study was compliant with the Helsinki and Oviedo declarations and approved by the Ethics Committee of IRCCS-INRCA Ancona (prot. n. 19699/20CE, date of approval 27 May 2020).

The principal investigator for each center was responsible for the collection, storage, and transmission of study data.

The study protocol was designed and conducted to ensure adherence to the principles and procedures of good clinical practice and comply with the laws of the respective countries of the participating centers, as described in the following documents [13,14]. 

In this study, we sought to compare (retrospectively) the ability to discriminate patients testing positive or negative for SARS-CoV-2 on a nasopharyngeal swab by corona score [4,5,6] with a novel corona score named the “LUS Corona Score”, which includes lung ultrasound instead of CXRs. The demographic data and the laboratory parameters previously considered in the corona score [4,5,6] were the same.

Point-of-care LUS was performed on patient admission to the emergency department, along with a physical examination, by emergency physicians trained in LUS. LUS and a physical examination were completed prior to CXR results (flowchart in Figure 1).

Institutions used similar methodologies for LUS scanning [7,8,15]. The interpretation of the exams was performed blindly by the physicians.

### 2.2. Measures and Equipment

LUS examinations were performed using one of the following three machines: an Esaote MyLab 25 (Esaote, Genova, Italy) [16], a Philips CX50 Philips, Amsterdam, Netherlands) [17], or a GE Healthcare Logiq E (GE Healthcare, LittleChalfont, UK [18]. In particular, an LA522 high-frequency linear array transducer with a bandwidth of 9–3 MHz and a CA430 convex array transducer with a bandwidth of 8–1 MHz were used with the Esaote MyLab 25, an L12-3 linear array transducer with a bandwidth of 12–3 MHz was used with the Philips CX5, and a linear array transducer with a bandwidth of 11–3 MHz was used with the GE Logiq with a lung preset. Scans were performed using longitudinal transducer orientation of the lung regions. 

X-rays were performed using both portable and fixed standard radiology equipment (Mecall Eidos RF 439) [19] and digitally transmitted via picture archiving and a communication system. Emergency physicians performing lung ultrasound examinations all had emergency ultrasound training, including lung ultrasound and hospital credentials. Each physician had performed at least 100 lung ultrasound examinations prior to the start of the study. Radiologists interpreting X-rays were all chest radiology specialists with hospital credentials in chest X-ray interpretation and were board-certified in diagnostic radiology.

Venous blood samples for laboratory analyses and nasopharyngeal swabs were performed when patients were admitted to the ED. Clinical chemistry analyses included C-reactive protein (CRP), ferritin, lactate dehydrogenase (LDH), absolute lymphocyte count (ALC), and absolute neutrophil count (ANC). The diagnosis of SARS-CoV-2 infection was confirmed using a real-time (rt) reverse transcriptase (RT) polymerase chain reaction (PCR) test for the qualitative detection of nucleic acid from SARS-CoV-2 (Abbott real-time SARS-CoV-2 RNA assay) [20]. 

### 2.3. Data Collection

Laboratory and instrumental data were collected in subjects tested for SARS-CoV-2 infection. CXRs and LUS were the exams routinely carried out during the usual time spent by patients in the emergency room. The corona score was calculated with laboratory measurements (CRP, ALC, ANC, LDH, and ferritin), age, sex, and CXRs as the inputs using the algorithm published [6] and validated [4] elsewhere.

Age: 0, 1, or 2 points

      Zero points for ages ≤ 75 years

      1 point for ages between 76 and 79 years

      2 points for ages ≥ 80; female: zero points

Sex: 0 or 1 point

      Zero points for women

      1 point for men

CRP: 0, 1, 2, or 3 points

      Zero points for CRP ≤ 9 mg/L

      1 point for CRP between 10 and 14 mg/L

      2 points for CRP between 15 and 38 mg/L

      3 points for CRP between 39 and 69 mg/L

      2 points for CRP between 70 and 193 mg/L

      1 point for CRP between 194 and 303 mg/L

      Zero points for CRP ≥ 304 mg/L

Ferritin: −1, 0, 1, 2, or 3 points

      −1 point for ferritin ≤ 15 μg/L

      Zero points for ferritin between 16 and 179 μg/L

      1 point for ferritin between 180 and 301 μg/L

      2 points for ferritin between 302 and 538 μg/L

      3 points for ferritin ≥ 539 μg/L

LDH: 0, 1, 2, or 3 points

      Zero points for LDH ≤ 257 U/L

      1 point for LDH between 258 and 265 U/L

      2 points for LDH between 266 and 397 U/L

      3 points for LDH ≥ 398 U/L

ALC: 0 or 1 point

      Zero points for ALC ≥ 1.3 × 10^9^/L

      1 point for ALC ≤ 1.2 × 10^9^/L

ANC: 0, −1, −2, −3, or −4 points

      −4 points for ANC ≥ 10.4 × 10^9^/L

      −3 points for ANC between 9.1 and 10.3 × 10^9^/L

      −2 points for ANC between 8 and 9 × 10^9^/L

      −1 point for ANC between 5.2 and 7.9 × 10^9^/L

      Zero points for ANC ≤ 5.1 × 10^9^/L

Infiltrates at CXR: 0, 1, or 4 points:

      Zero points for no infiltrate

      1 point for unilateral infiltrate

      4 points for bilateral infiltrate

We calculated two modified corona scores: the modified 1 corona score (M1CS) and modified 2 corona score (M2CS).

In the M1CS, we integrated the corona score with the findings of interstitial pattern at CXRs and pleural irregularities at LUS. Summarizing, the M1CS considers the genuine corona score by adding the following:

Interstitial pattern at the CRX: 0 or 2 points

      Zero points for no interstitial pattern at the CRX

      2 points for an interstitial pattern at the CRX

Irregular pleural line at LUS: 0 or −1 point

      Zero points for no irregular pleural line at LUS

      −1 point for an irregular pleural line at LUS

In the M2CS, we excluded all CXR findings and considered only the following items:

Age: 0, 1, or 2 points

      Zero points for ages ≤ 75 years

      1 point for ages between 76 and 79 years

      2 points for ages ≥ 80; female: zero points

Sex: 0 or 1 point

      Zero points for women

      1 point for men

CRP: 0, 1, 2, or 3 points

      Zero points for CRP ≤ 9 mg/L

      1 point for CRP between 10 and 14 mg/L

      2 points for CRP between 15 and 38 mg/L

      3 points for CRP between 39 and 69 mg/L

      2 points for CRP between 70 and 193 mg/L

      1 point for CRP between 194 and 303 mg/L

      Zero points for CRP ≥ 304 mg/L

Ferritin: −1, 0, 1, 2, or 3 points

      −1 point for ferritin ≤ 15 μg/L

      Zero points for ferritin between 16 and 179 μg/L

      1 point for ferritin between 180 and 301 μg/L

      2 points for ferritin between 302 and 538 μg/L

      3 points for ferritin ≥ 539 μg/L

LDH: 0, 1, 2, or 3 points

      Zero points for LDH ≤ 257 U/L

      1 point for LDH between 258 and 265 U/L

      2 points for LDH between 266 and 397 U/L

      3 points for LDH ≥ 398 U/L

ALC: 0 or 1 point

      Zero points for ALC ≥ 1.3 × 10^9^/L

      1 point for ALC ≤ 1.2 × 10^9^/L

ANC: 0, −1, −2, −3, or −4 points

      −4 points for ANC ≥ 10.4 × 10^9^/L

      −3 points for ANC between 9.1 and 10.3 × 10^9^/L

      −2 points for ANC between 8 and 9 × 10^9^/L

      −1 point for ANC between 5.2 and 7.9 × 10^9^/L

      Zero points for ANC ≤ 5.1 × 10^9^/L

Irregular pleural line at LUS: 0 or −1 point

      Zero points for no irregular pleural line at LUS

      −1 point for an irregular pleural line at LUS

### 2.4. Statistical Analysis

All 265 subjects considered in our study were divided into two groups by the result of the molecular test for SARS-CoV-2. Full clinical data were available in 88% and 92% of positive and negative subjects, respectively, and most missing variables were for smoking status, heart rate, and blood pressure. Categorical (qualitative) variables were gender, symptoms, hemodynamic instability, and medical history of dementia, diabetes, neoplasms, or lung or heart diseases. 

Categorical variables were presented as percentages, whereas continuous variables were illustrated as median ± standard error. Differences between positive and negative subjects on molecular testing were examined using *t*-tests and χ^2^ tests as appropriate. Correlations between the molecular test and clinical variables (CXR and LUS) were explored by means of multiple regression after adjusting for the corona score, blood pressure, oxygen saturation, and medical history using Statistica 6.0 (StatSoft Inc, Tulsa, OK, USA). ROC curves were plotted using MedCalc v 22.106 (MedCalc Software Ltd., Ostend, Belgium). Sensitivity and specificity of the corona score, M1CS, and M2CS were estimated as the area under the ROC curve.

## 3. Results

A total of 265 patients were considered in this retrospective study. Table 1 illustrates the subjects’ demographic, laboratory, historical, and physical findings. Data are shown as mean ± standard error for positive and negative subjects, respectively. The mean age was 65.5 ± 2.7 and 71.6 ± 1.2 years for the two groups; females were 53.1% and 44.0%, respectively.

Subjects positive on molecular nasal swab testing were older and had a similar sex distribution. They presented higher ferritin levels and similar lactate dehydrogenase, absolute lymphocyte, and neutrophil counts. Moreover, they showed higher temperatures without differences in cough, asthenia, gastroenterological symptoms, hemoptysis, arthromyalgia, chest pain, ageusia, anosmia, or other symptoms. They had a similar systolic blood pressure and heart rate and a lower diastolic blood pressure and oxygen saturation. They were also more likely to have a history of dementia and less likely to have a history of cancer, but they did not differ in their history of diabetes or lung or heart diseases. 

As reported in Table 2, the two groups differed on the corona score and modified corona scores (*p* < 0.001 for all). Subjects positive at the molecular swab had fewer pleural line irregularities (*p* = 0.025) and more interstitial patterns on CXRs (*p* < 0.001). 

In the two groups, comparable LUS findings were pleural effusions, isolated and confluent B lines, and small and big consolidation areas (Figure 2). 

In the same groups, similar CXR findings were normal findings, ground glass, and infiltrates (Table 2). The area under the ROC curve for the corona score was 0.713 (95% CI 0.601–0.822) (Figure 3).

The M1CS, including LUS findings, as well as the CXR pattern, resulted in slightly higher sensitivity and specificity, with an area under the ROC curve of 0.758 (95% CI 0.649–0.858), as shown in Figure 4.

In our study, the M2CS, which included LUS findings instead of CXR results, yielded even better sensitivity and specificity, with an area under the ROC curve of 0.771 (95% CI 0.673–0.869). The sensitivity and specificity of the M1CS and M2CS are reported in Table 3.

## 4. Discussion

We previously reported on the corona score, incorporating multiple demographic and laboratory data points along with CXR and LUS findings. However, CXRs and more advanced imaging modalities may not be available in some healthcare locations, such as nursing homes, private homes, many clinics, and similar facilities. Additionally, large areas in developing countries may not have reliable electricity or funding for CXRs, leaving a portable ultrasound as the only practical imaging modality. 

Interestingly Ord and Griksaitis have repeatedly shown that LUS has higher sensitivity and specificity than CXRs in diagnosing consolidations, pleural effusion, and interstitial syndrome, which are typical in pneumonia [21]. LUS has also been shown to be an important diagnostic tool in epidemic and pandemic respiratory infections [22,23]. A decade prior to the current COVID-19 pandemic, a group of authors demonstrated how bedside lung ultrasound was useful during a pandemic event, providing early detection of interstitial involvement in influenza A (H1N1)v pneumonia, even when the CXR was normal [24].

Thus, the use of LUS as the sole imaging modality, coupled with laboratory results and demographic factors, may be an acceptable diagnostic path compared to the incorporation of traditional imaging modalities. The development of an easy-to-perform corona score with ultrasound and other basic data points can facilitate care in many out-of-hospital and low-resourced environments. 

In our study, we tested the possibility of integrating LUS results into the corona score in order to enhance its usefulness and obtain a better predictive value. Our goal was to support safely limiting the amount of blood tests performed and the systematic and indiscriminate use of higher-level imaging methods, like computed tomography (CT). These more invasive and expensive imaging methods were frequently overused, as evidenced in the course of the first wave of the pandemic and, were previously highlighted in a review of the literature [25,26]. The M1CS compared to the traditional corona score showed slightly improved sensitivity and specificity.

Our second goal, which was more ambitious, was to explore replacing the use of the CXR included in the traditional corona score with LUS and compare the results obtained with the previous score. We called this the hypothetical new modified corona score, M2CS. Replacing CXRs with LUS yielded satisfactory results and showed better sensitivity and specificity than the M1CS. 

Replacing CXRs with LUS in the corona score assessment enables its use in remote areas or where reliable swab testing for SARS-CoV-2 is not immediately available. This may be especially meaningful when considering that the more widely available, cheaper, and faster antigen tests sometimes give false positive or negative results. Equally important and especially relevant in many developed countries is the usefulness of a modified score with LUS in residences for the elderly or in similar locations that do not have access to CT and X-rays.

Lung ultrasound has developed into an ideal imaging tool for the evaluation of COVID-19-infected patients across a broad swatch of medical settings. Its use is even supported by large traditional imaging medical specialties due to contamination and radiation concerns from the overuse of chest CT for COVID-19 patients. 

Another important reason to support the use of LUS, instead of CXRs, in the corona score for patients with suspected COVID-19 is to reduce the risk associated with transporting unstable patients to radiology. This would further act to reduce nosocomial outbreaks due to this highly contagious virus [27]. A recent study underlined the concept that modern radiology rooms have become a “meeting point” where it is all too easy for cross-infection to occur. Furthermore, there is some indication that radiology staff may lack the knowledge and training to effectively implement infection control practices [28].

Since the start of the pandemic in China over three years ago, multiple algorithms have been published describing the use of lung ultrasound, alone or in combination with other testing, to risk stratify COVID-19 patients with lung involvement. However, few studies have compared different risk stratification or scoring approaches, and many simply described single-center or single-approach methodology.

The application of traditional lung ultrasound approaches, which are heavily focused on pleural reverberation artifacts, commonly termed B lines, often resulted in inadequate specificity due to the frequent finding of B lines across a variety of infectious and non-infectious disease states. A frequently confounding scenario occurred when patients with possible cardiac pulmonary edema now had included COVID-19 infection in their differential diagnosis. Reliance on B lines led to confusion and inability to separate pulmonary edema and other chronic lung conditions from COVID-19 infection. 

In time, astute LUS practitioners realized that COVID-19 presented significant pleural inflammatory effects, which were rarely seen in other processes, especially when combined with typical distribution patterns in infected patients. These findings were first documented on chest CT, even before the first LUS for COVID-19 manuscripts was published. The importance of pleural thickening in COVID-19 diagnosis and risk stratification was more broadly realized and began to be incorporated into risk and prediction scores. One of the starkest examples of pleural line importance was delivered in a LUG and machine learning study designed to differentiate hydrostatic pulmonary edema, ARDS, and COVID-19 lung infection among ICU patients. Arntfield et al. found that their algorithm was far superior to human LUS experts at differentiating among the three processes, and a deeper analysis revealed the key was focusing on the pleural line [29]. An analysis of key factors behind the algorithm’s success showed the importance of pleural line morphology analysis.

LUS seemed to perform well when coupled with laboratory results and functional assessments, such as at-rest pulse oximetry and the effect of walking on patient oxygen saturations. In fact, several studies demonstrated the effectiveness of LUS versus different methods. Bass et al. studied the possibility of using LUS and oximetry in patients with acute respiratory distress syndrome (ARDS), especially in scarce resource scenarios. This prospective study, performed in patients with acute respiratory failure undergoing mechanical ventilation, demonstrated that the use of point-of-care LUS together with pulse oximetry provides reasonable sensitivity in identifying patients with ARDS [30].

Carlucci et al. performed a study in patients with previous SARS-CoV-2 pneumonia who had recovered and did not show blood gas alteration. They highlighted the usefulness of LUS in selecting patients to undergo a standardized test, like a 6 min walking test (6MWT), to identify those at risk of having exercise-induced desaturation (EID) who would be candidates for a rehabilitation program [31]. Another recent study further underlined the indisputable importance of the use of LUS for the diagnosis of COVID-19 [32].

Our study has several weaknesses because it is a retrospective study and we had access to static ultrasound images. This limit resulted from a lack of routine storage of video cine lops during patient examination, which prevented us from performing an inter-reviewer agreement analysis to strengthen the results. We suggest that all providers have the capability to store video cine loops along with static images in order to allow for a more thorough retrospective analysis of patient findings.

This study found that the new corona scores, including LUS findings instead of CXRs, are not inferior to the original one.

## 5. Conclusions

Lung ultrasound compared with other modalities, such as chest X-rays, can be used at home or in healthcare facilities [20] where imaging services, except for LUS, are not readily available. LUS may be more easily used and incorporated into the corona score instead of chest X-rays.

## Figures and Tables

**Figure 1 diagnostics-14-00051-f001:**
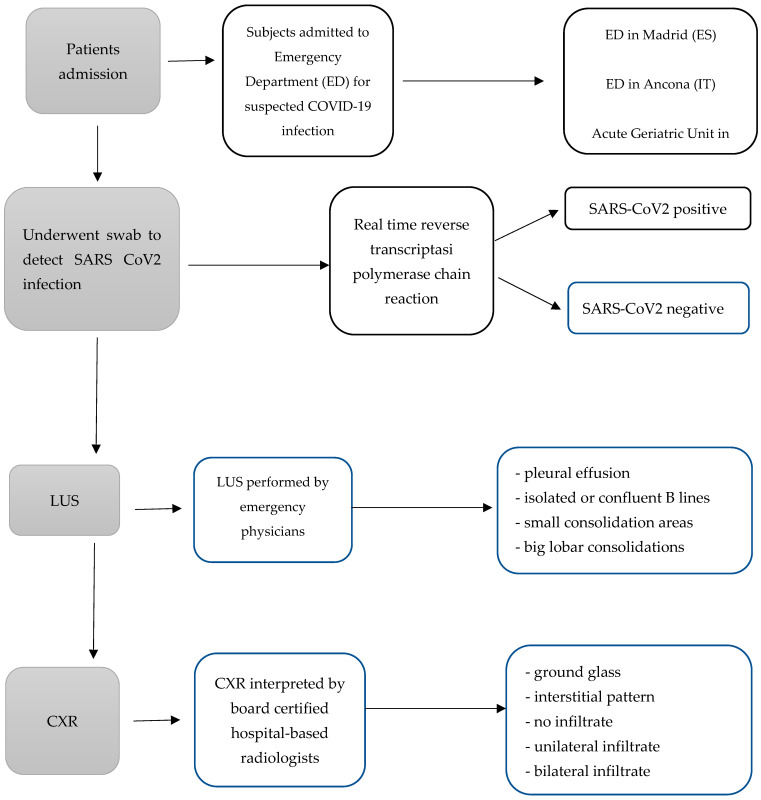
The diagnostic pathways carried out in subjects included in the retrospective study.

**Figure 2 diagnostics-14-00051-f002:**
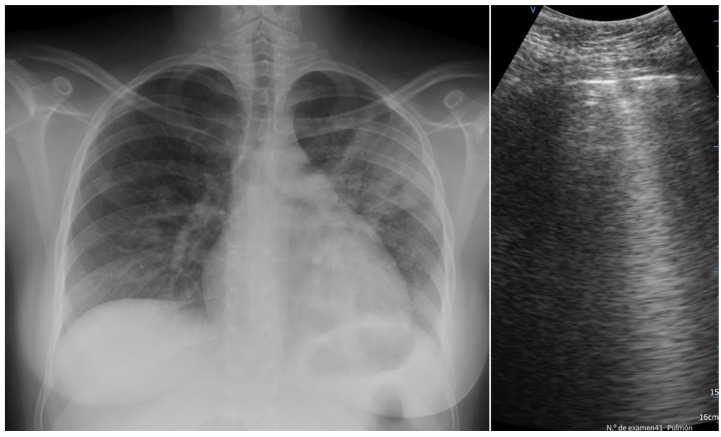
A CXR and LUS of the same subject, respectively, with bilateral infiltrates and small consolidation areas.

**Figure 3 diagnostics-14-00051-f003:**
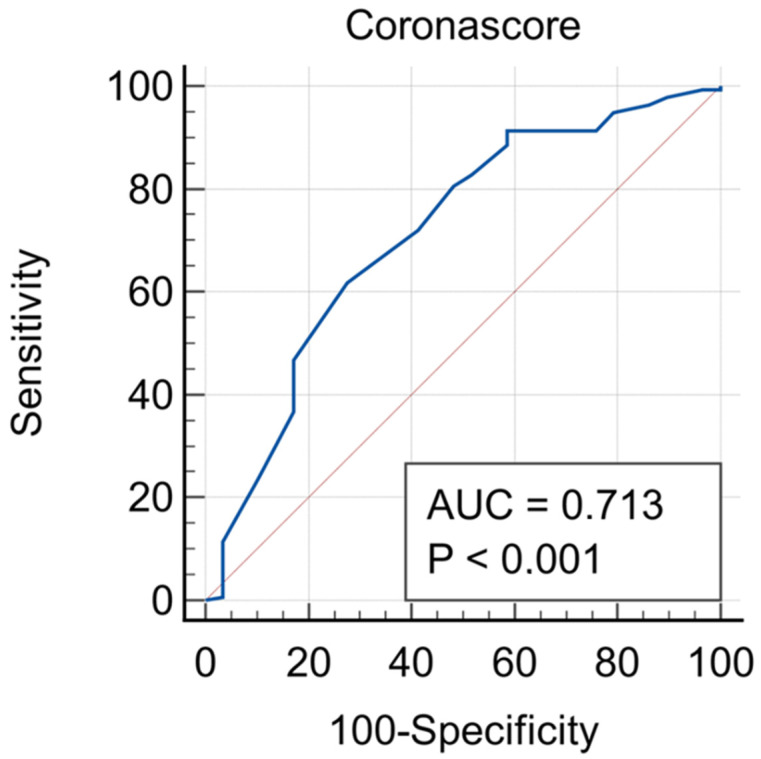
Sensitivity and specificity of a genuine corona score. The area under the ROC curve was 0.713 (95% CI 0.601–0.822) for the score.

**Figure 4 diagnostics-14-00051-f004:**
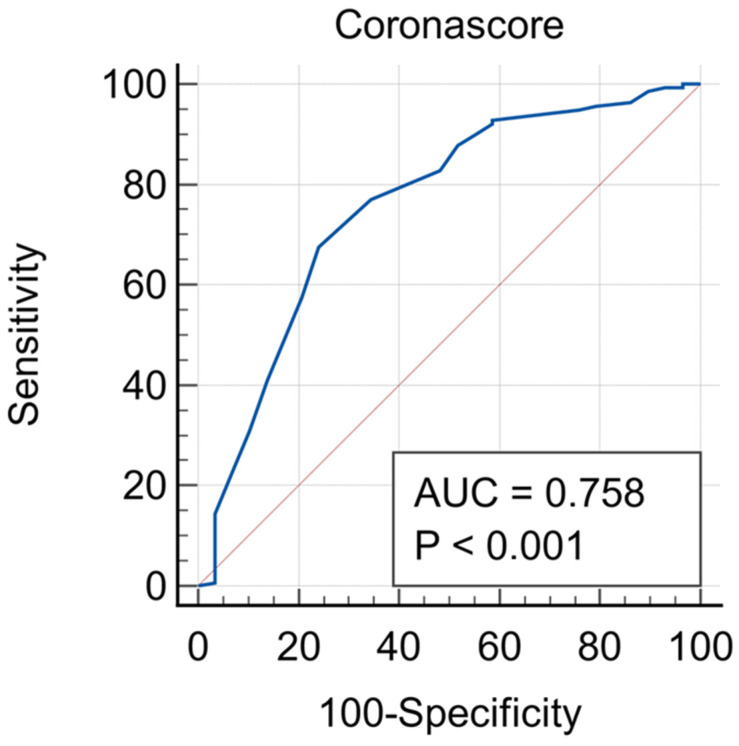
Sensitivity and specificity of the modified 1 corona score (M1CS). The area under the ROC curve was 0.758 (95% CI 0.649–0.858) for the M1CS.

**Table 1 diagnostics-14-00051-t001:** Clinical characteristics of the sample by swab.

	Negative on the Molecular Test (*n* = 49)	Positive on the Molecular Test (*n* = 216)	*p*
Age (years)	65.5 ± 2.7	71.6 ± 1.2	0.03
Female	53.1%	44.00%	ns
C-reactive protein (mg/dL)	50.6 ± 11.1	53.7 ± 5.8	ns
Ferritin (ug/L)	492.5 ± 85	975.9 ± 68.2	<0.01
Lactate dehydrogenase (U/L)	329.8 ± 41.5	415.1 ± 19.1	ns
Absolute lymphocyte count (10^9^/L)	1.2 ± 0.2	1.2 ± 0.1	ns
Absolute neutrophil count (10^9^/L)	6.2 ± 0.5	5.1 ± 0.4	ns
Fever	46.7%	75.0%	<0.01
Dyspnoea/Cough	69.4%	70.4%	ns
Asthenia	38.8%	26.9%	ns
Gastroenterological symptoms	10.2%	15.3%	ns
Hemoptysis	0.0%	2.8%	ns
Arthromyalgia	0.0%	4.6%	ns
Chest pain	14.3%	16.7%	ns
Ageusia/Anosmia	2.0%	3.7%	ns
Bradycardia	0.0%	1.9%	ns
Faringodinia	0.0%	2.4%	ns
Cefalea	0.0%	1.9%	ns
Systolic blood pressure (mmHg)	130.6 ± 3.4	122.8 ± 1.7	ns
Diastolic blood pressure (mmHg)	76.1 ± 2.1	70.7 ± 1.0	0.02
Heart rate (bpm)	94.0 ± 2.4	90.2 ± 1.3	ns
Oxygen saturation	93.1 ± 0.9	90.9 ± 0.4	0.02
Hemodynamic instability	4.1%	14.2%	ns
History of dementia	12.2%	45.8%	<0.01
History of diabetes	34.7%	35.6%	ns
History of lung disease	28.6%	25.9%	ns
History of neoplastic disease	28.6%	10.2%	<0.01
History of heart disease	28.0%	22.7%	ns

Mean ± standard error. Differences in significance were analyzed using a *t*-test or χ^2^, as appropriate. ns: not significant.

**Table 2 diagnostics-14-00051-t002:** Corona score, modified corona scores (M1CS and M2CS), and imaging (ultrasound and X-ray) of the lung’s findings overall and with a swab.

	Positive at Swab	Negative at Swab	*p*
Corona score	9.8 ± 0.3	6.7 ± 0.8	<0.001
Modified 1 corona score: M1CS	10.5 ± 0.3	6.4 ± 0.9	<0.001
Modified 2 corona score: M2CS	6.4 ± 0.2	3.5 ± 0.5	<0.001
LUS pleural effusion	13.9%	20.4%	ns
LUS isolated B lines	81.0%	81.6%	ns
LUS confluent B lines	44.0%	55.1%	ns
LUS irregular pleural line	67.6%	91.8%	0.025
LUS small consolidation areas	64.8%	57.1%	ns
LUS big/lobar consolidation	13.0%	10.2%	ns
X-ray normal findings	17.0%	26.5%	ns
X-ray ground glass	35.6%	35.5%	ns
X-ray interstitial pattern	61.9%	29.4%	<0.001
X-ray no infiltrate	27.2%	32.3%	ns
X-ray unilateral infiltrate	23.3%	11.8%	ns
X-ray bilateral infiltrate	49.5%	55.9%	ns

Mean ± standard error. Differences in significance were analyzed using multiple regressions or χ^2^, as appropriate. ns: not significant.

**Table 3 diagnostics-14-00051-t003:** Sensitivity and specificity of the M1CS and M2CS (95% CI).

Corona Score Cut-Off Value	M1CS		M2CS	
	Sensitivity	Specificity	Sensitivity	Specificity
2	98.4% (0.95–0.99)	8.2% (0.03–0.20)	92.6% (0.89–0.96)	20.4% (0.11–0.35)
3	96.3% (0.93–0.98)	12.4% (0.05–0.25)	91.1% (0.83–0.97)	30.6% (0.19–0.45)
4	95.8% (0.93–0.98)	24.5% (0.14–0.39)	85.7% (0.80–0.91)	49.0% (0.34–0.64)
5	87.7% (0.82–0.92)	28.6% (0.16–0.44)	73.7% (0.67–0.80)	59.2% (0.44–0.73)
6	85.7% (0.79–0.90)	32.7% (0.20–0.48)	62.6% (0.55–0.69)	69.4% (0.55–0.82)
7	80.9% (0.73–0.85)	36.8% (0.23–0.52)	49.3% (0.42–0.55)	75.5% (0.61–0.84)
8	71.1% (0.64–0.77)	49.9% (0.35–0.64)	36.3% (0.30–0.44)	85.7% (0.73–0.94)
9	62.1% (0.55–0.69)	57.8% (0.43–0.72)	17.2% (0.12–0.23)	96.0% (0.86–0.99)
10	52.9% (0.46–0.60)	70.8% (0.55–0.82)	8.9% (0.05–0.14)	98.0% (0.90–0.99)

## Data Availability

All data generated or analyzed in this study are included in this published article.

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
