# Peer review of "A Modified Corona Score Using Lung Ultrasound to Identify COVID-19 Patients"

_diagnostics, 2023, doi:10.3390/diagnostics14010051_

Round 1

Reviewer 1 Report

Comments and Suggestions for Authors

Nicely composed manuscript. Here are some small changes the authors are encouraged to consider:

1) The abstract can be shortened to include the most appropriate content.

2) Figure 1 can be rearranged to enhance neat ness

3) Figure 4 needs better explanation

4) Literature on recent development in ultrasound should be cited:

iii) "The use of lung ultrasound in COVID-19": DOI: 10.1183/23120541.00196-2022

Comments on the Quality of English Language

Not applicable

Author Response

Reviewer 1

Thank you for your suggestions. Changes are highlighted in yellow

Q1: The abstract can be shortened to include the most appropriate content.

A1: I shortened the abstract to include the most appropriate content.

Q2: Figure 1 can be rearranged to enhance neat ness

A2: I rearranged Figure 1

Q3: Figure 4 needs better explanation

A3: I explaned better Figure 4

Q4:Literature on recent developmet in ultrasound should be cited: “the use of lung ultrasound in COVID-19”; DOI: 10.1183/23120541.00196-2022

A4: I cited this paper, like you suggested

Reviewer 2 Report

Comments and Suggestions for Authors

November 6, 2023

To: Diagnostics MDPI

Dear EIC,

Dear AE,

I hope you are doing well.

This is my review report for the manuscript ID: diagnostics-2703431.

This study evaluated the diagnostic accuracy and power of two different clinical methods (LUS and CXR) by scoring and applying the valuable test diagnostic accuracy criteria, ROC curves and AUC. I think this study can add some novel things to the field. However. I’ve prepared several minor and major comments to level up the manuscript.

Comments:

·      The introduction should be truncated to less than one page.

·      The manuscript is well-written however, I noticed some errors in this regard. For example, see the line 178; “were studied” is correct. Also, I saw some incorrect spaces between words.

·      At the statistical analysis section, what kinds of softwares were applied to analyses, especially ROC curves?

·      At the statistical analysis section: the authors should define how many variables were analyzed? How many and what variables were qualitative and how many and what variables were quantitative?

·      Kindly give valuable and more references for material method. For example, I didn’t see any refs in 2.2 section.

·      The results section is not adequate. It should be reached at least to one page. I suggest to expand the finding which have abstracted in the tables.

·      Kindly expand the main findings of Figures 2, 3 in their legends.

·      The difference between the AUC rates are not adequate to claim that LUS was superior to CXR. The authors should discuss this. Also, I think this claim should be carefully considered.

Author Response

Reviewer 2

Thank you for your suggestions. Changes are highlighted in yellow

Q1: The introduction should be truncated  to less than one page

A1: I truncated the introduction

Q2: The manuscript is well-written however, I noticed some errors in this regard. For example, see the line 178;  “’were  studied”  is correct. Also I saw some incorrect paces between words.

A2: I corrected the errors

Q3: At the statistical analysis section, what kinds of software were applied to analyses, especially ROC curves?

A3: Thanks for your question. Data were analyzed data by means of Statistica 6.0 (StatSoft Inc, Tulsa, Oklahoma, USA). ROC curves were plotted using MedCalc v22.106 (MedCalc Software Ltd, Ostend, Belgium). Both software applications are now cited in the manuscript (lines 295-297).

Q4: At the statistical analysis section: the authors should define how many variables were analyzed? How many and what variables were qualitative and how many and what variables were quantitative?

A4: Thanks for your question and sorry for our lacking. Categorical variables are now all clearly stated (lines 287-288). We adjusted multiple regressions for genuine Corona-score, blood pressure, oxygen saturation and medical history as described (lines 293-295)

Q5: Kindly give valuable and more references for material method. For example, I didn’t see any refs in 2.2 section.

A5:  I added some appropriate references in material method section

Q6: The results section is not adequate. It should be reached at least to one page. I suggest to expand the finding which have abstracted in the tables.

A6: Following your suggestion, more results are now reported in the text from tables (Lines 307-315). Moreover, Table 3 is now added, as kindly requested by other reviewers.

Q7: kindly expand the main findings of figure 2, 3 in their legend

A7: I expanded the main findings of figure 2 and 3

Q8: The difference between the AUC rates are not adequate to claim that LUS was superior to CXR. The authors should discuss this. Also, I think this claim should be carefully considered.

A8: Thanks for your deepening which could be very useful for our next studies. Our work was not designed to claim that M1CS and M2CS are superior to genuine Corona-score. We tried to compare M1CS and M2CS to genuine Corona-score in terms of AUC finding that modified corona scores are not inferior to the previous one (Line 447-448).

Reviewer 3 Report

Comments and Suggestions for Authors

Prospective studies are more likely to answer clinical questions than retrospective studies as the letter are prone to have missing data, and are more likely to have biases. It is thus important that in this retrospective study, information is provided about how the patients' data were obtained, and in what proportion of patients there were full clinical data. 

The paper should have more detail about how the scores were calculated so that clinicians who wish to use one of these scoring systems should be able to easily apply it in their practices.

If the scores are to be used for making the diagnosis of Covid-19, then knowing the specificities for different cut-offs is important.  If the scores are to be used for screening, then scores with high sensitivities are important to be sure that patients can be reassured that they do not have Covid-19. Thus, the paper should have tables of specificities and sensitivities of different cut-offs for these scores.

Author Response

Reviewer 3

Thank you for your suggestions. Changes are highlighted in yellow

Q1: Prospective studies are more likely to answer clinical questions than retrospective studies as the letter are prone to have missing data, and are more likely to have biases. It is thus important that in this retrospective study, information is provided about how the patients' data were obtained, and in what proportion of patients there were full clinical data.

A1: Thanks for your this question. Data were collected from medical records. The information about missing data are now in the manuscript (Lines 284-286).

Q2: The paper should have more detail about how the scores were calculated so that clinicians who wish to use one of these scoring systems should be able to easily apply it in their practices.

A2: Thanks for your suggestion. Variables used for the 3 scoring systems are now clearly stated in the text (Lines 188-278).

Q3: If the scores are to be used for making the diagnosis of Covid-19, then knowing the specificities for different cut-offs is important.  If the scores are to be used for screening, then scores with high sensitivities are important to be sure that patients can be reassured that they do not have Covid-19. Thus, the paper should have tables of specificities and sensitivities of different cut-offs for these scores.

A3: Thanks for your deepening about cut-offs and statistics and sorry for our lacking. Following your suggestion, Tables 3 is now added showing sensitivity and specificity of M1CS and M2CS.